# A Comprehensive Review of Pediatric Acute Encephalopathy

**DOI:** 10.3390/jcm11195921

**Published:** 2022-10-07

**Authors:** George Imataka, Shigeko Kuwashima, Shigemi Yoshihara

**Affiliations:** 1Department of Pediatrics, Dokkyo Medical University, Tochigi 321-0293, Japan; 2Department of Radiology, Dokkyo Medical University, Tochigi 321-0293, Japan

**Keywords:** acute encephalopathy, convulsions, pediatrics, brain hypothermia

## Abstract

Acute encephalopathy typically affects previously healthy children and often results in death or severe neurological sequelae. Acute encephalopathy is a group of multiple syndromes characterized by various clinical symptoms, such as loss of consciousness, motor and sensory impairments, and status convulsions. However, there is not only localized encephalopathy but also progression from localized to secondary extensive encephalopathy and to encephalopathy, resulting in a heterogeneous clinical picture. Acute encephalopathy diagnosis has advanced over the years as a result of various causes such as infections, epilepsy, cerebrovascular disorders, electrolyte abnormalities, and medication use, and new types of acute encephalopathies have been identified. In recent years, various tools, including neuroradiological diagnosis, have been developed as methods for analyzing heterogeneous acute encephalopathy. Encephalopathy caused by genetic abnormalities such as CPT2 and SCN1A is also being studied. Researchers were able not only to classify acute encephalopathy from image diagnosis to typology by adjusting the diffusion-weighted imaging/ADC value in magnetic resonance imaging diffusion-weighted images but also fully comprehend the pathogenesis of vascular and cellular edema. Acute encephalopathy is known as a very devastating disease both medically and socially because there are many cases where lifesaving is sometimes difficult. The overall picture of childhood acute encephalopathy is becoming clearer with the emergence of the new acute encephalopathies. Treatment methods such as steroid pulse therapy, immunotherapy, brain hypothermia, and temperature control therapy have also advanced. Acute encephalopathy in children is the result of our predecessor’s zealous pursuit of knowledge. It is reasonable to say that it is a field that has advanced dramatically over the years. We would like to provide a comprehensive review of a pediatric acute encephalopathy, highlighting advancements in diagnosis and treatment based on changing disease classification scenarios from the most recent clinical data.

## 1. Introduction

Acute encephalopathy in childhood and adolescence refers to a brain pathobiological condition that progresses rapidly. Acute encephalopathy is a syndrome that is characterized by central nervous system dysfunction caused by diffuse or widespread noninflammatory cerebral edema [1]. A task force of experts from ten academic societies recently developed a consensus-based, uniform nomenclature for acute cognition disturbances and defined it as a rapidly developing pathophysiological brain process manifesting as subsyndromal delirium, delirium, or coma [1]. This position statement defines subsyndromal delirium as a state intermediate between normal cognition and delirium in which none of the Diagnostic and Statistical Manual of Mental Disorders, Fifth Edition (DSM-5) criteria for delirium is met. The Japanese Society of Child Neurology guidelines for acute encephalopathy in infancy and childhood propose that the impairment of consciousness in acute encephalopathy persists for at least 24 h with 11 points or less on the Glasgow coma scale [2]. Some encephalopathy disorders are multifactorial, whereas others, such as previous viral infection or hepatic or uremic encephalopathy have a known etiology. In Japan, influenza virus is the leading cause of mild encephalitis/reversible splenic lesion (MERS) and acute necrotizing encephalopathy (ANE). HHV-6 is the most common cause of AESD (acute encephalopathy with biphasic seizures and reduced diffusion). Bacteria and mycoplasma each accounted for 2% and 1% of the total. MERS and ANE are the most common syndromes, followed by AESD [3,4]. In recent years, there has been an increase in the number of cases of acute encephalopathy associated with other viruses, such as human metapneumovirus, rhinovirus, and cytomegalovirus [4].

Acute encephalopathy is fairly common in East Asia. From spring 2007 to spring 2010, Hoshino et al. conducted a pediatric acute encephalopathy survey in Japan, recording 983 cases of pediatric acute encephalopathy over a 3-year period. The questionnaire collection rate was approximately 50%, and the annual number of pediatric acute encephalopathy cases in Japan is estimated to be 400 to 700. Among acute encephalopathy syndrome, AESD was the most common (29%), MERS was the second most common (16%), and ANE was the least common (4%). Other syndromes that occurred in 2% of cases included hemorrhagic shock encephalopathy syndrome (HSES) [3]. The annual incidence in Japan had increased to 1115 cases by 2017 [4]. Despite the fact that the annual incidence rate in Japan is about 1000, it is significantly greater than the number of cases of acute encephalopathy in Western countries. According to a statewide questionnaire survey conducted in Japan in 2012, 56.2% of those who suffered from acute encephalopathy recovered completely, 22.1% had mild to moderate sequelae, 13.5% had severe sequelae, and 5.6% died [3]. The total percentage of acute encephalopathy-related deaths in Japan is 6%, with 36% developing neurologic sequelae. Deaths due to AESD are uncommon, but neurologic complications are common. In ANE and HSES, both fatalities and neurologic sequelae are common [2].

Despite the relatively high morbidity and mortality associated with acute encephalopathy, there is limited evidence for diagnosing and treating acute encephalopathy. Virus epidemics, regional differences, racial disparities, human leukocyte antigen (HLA) differences, and medical level challenges are all discussed, but the true cause of the high incidence of acute encephalopathy in Japanese children remains unknown. Furthermore, various conditions such as intracranial infections, autoimmune encephalitis, cerebrovascular diseases, traumatic, metabolic and toxic disorders, and the effects of organ failure manifest with acute impairment of consciousness and may complicate the diagnosis and management of acute encephalopathy. Assessing and comprehending current clinical practice patterns and variations is an important first step in promoting future research to develop evidence-based management and, ultimately, reduce the morbidity and mortality associated with acute encephalopathy. The purpose of this review is to offer an updated summary of the available data and a perspective on the diagnosis and management of pediatric acute encephalopathy. A detailed analysis of the epidemiological, clinical presentation, diagnosis, and management of pediatric acute encephalopathy was beyond the scope of the authors.

## 2. Classification of Acute Encephalopathy and Literature Search

We conducted a literature search from 1 January 1966 to 1 September 2022, using the MEDLINE/PubMed and National Institutes of Health Clinical Trials Registry (http://www.clinicaltrials.gov (accessed on 2 June 2022)) electronic medical databases for the identification of publications on acute encephalopathy articles were included if they were published in the English language. We excluded conference posters. Keywords included acute encephalopathy, pediatric, children, preschool, newborn, infant, acute febrile encephalopathy, and status epilepticus. Abstracts and full-text articles of randomized clinical trials, reviews, and other study designs were considered from studies describing relevant data on pediatric acute encephalopathy. An additional search was carried out via Google Scholar, and relevant articles on prospective and retrospective designs and real-world data on pediatric acute encephalopathy were considered.

To date, over ten acute encephalopathy syndromes have been identified and described. They are classified into four categories based on their primary pathogenesis: metabolic error, cytokine storm, excitotoxicity, and unknown mechanisms (Table 1). Unclassifiable encephalopathies together with a combination of several pediatric acute encephalopathies such as the co-occurrence of AESD with MARS or ANE with MARS and ANE are named multiple encephalopathy syndrome (MES) [5,6,7,8,9,10,11,12,13,14,15]. Carnitine transpalmitoyl transferase II, Toll-like receptor 3 single nucleotide polymorphisms, adenosine A2A receptor, neuronal sodium channel alpha 1 subunit (SCN1A) and SCN2A mutation have all been identified as risk factors for acute encephalopathy [6,7,8,9,10]. Acute encephalopathy that develops in association with acute adrenal insufficiency (congenital adrenal hyperplasia (CGH]), caused by fever or gastroenteritis, has also been reported [16]. Epileptic encephalopathies are a set of epileptic disorders that manifests itself early in life. Early infantile epileptic encephalopathy with suppression bursts (EIEE/Ottawa syndrome) and early myoclonic encephalopathy (EME) both occur in early infancy [17]. West syndrome affects children between the ages of 4 months and 2 years and manifests as infantile spasms [18]. Lennox–Gastaut syndrome develops between the ages of 2 and 6 years [19]. In addition, infantile epilepsy with migrating focal seizures, EIMFS, which occurs at less than 6 months of age, is also significant. Some of these developmental epileptic encephalopathies are linked to STXBP1, ARX, and other proteins. Some patients have KCNT1, SCN1A, KCNQ2, SCN2A, SCN8A, TBC1D24, SLC25A22, and SLC12A5 genetic abnormalities [20]. Dravet syndrome (infantile severe myoclonic epilepsy SMEI) is caused by mutations in SCN1A, SCN1B, SCN2A, and GABRG2. Before the age of one year, the majority of Dravet syndrome patients experience recurrent or status febrile seizures [21].

## 3. Clinical Presentation

A pathologic feature of acute encephalopathy is noninflammatory brain edema. This pathologic feature increases intracranial pressure, which leads to decreased cerebral perfusion pressure and, eventually, herniation syndromes and/or brainstem dysfunction associated with central nervous system-caused respiratory and circulatory failure [19,27]. Seizures are common in many people, and they are often febrile and last for a long time (febrile status epilepticus). Depending on the child’s age, there may be a change in personality or behavior as well as a decrease in cognitive functioning, developmental regression/stasis, a reduction in conscious level, and specific localizing features such as seizures, ataxia, tremor, or other focal motor symptoms. Fever, vomiting, lethargy, loss of appetite, and headache are all examples of systemic symptoms. Regardless of the cause of encephalopathy, all cases of acute encephalopathy have at least one symptom, namely an altered mental state. The altered mental state can be subtle and develop over time, such as apraxia, or the inability to sketch simple drawings, or it can be obvious and develop quickly, leading to coma or death within minutes [28].

The clinical course of metabolic errors and inherited metabolic disorders may include gradually progressive or static features, followed by the emergence of an acute encephalopathic crisis, including lethargy, behavioral changes, or gait disturbances caused by infections or a fasting state. Patients presenting with a cytokine storm may have systematic inflammatory response syndrome, which includes (1) increased or depressed leukocytes or 10% immature neutrophils, (2) tachycardia or bradycardia, (3) tachypnea or the need for mechanical ventilation, and (4) elevated or depressed leukocytes or 10% immature neutrophils. Acute excitotoxic encephalopathy, a mild encephalopathy caused by excitotoxicity is defined as a loss of consciousness that lasts more than 24 h and is usually accompanied by seizures but does not have a biphasic clinical course. Conversely, AESD is clinically recognized by biphasic seizures; an early seizure that is a prolonged febrile seizure on day 1, followed by late seizures that are a cluster of complex partial seizures on days 4–6 [29].

## 4. Diagnosis

A coma with obvious consciousness impairment or a convulsive condition is a clinical indicator of acute encephalopathy; however, identifying acute encephalopathy in these circumstances is fairly easy. However, there are various early signs and symptoms as well as variations in these symptoms. This large range of clinical symptoms mirrors the wide range of cerebral function abnormalities as provided by the International Encephalitis Consortium, which recommends the diagnosis of encephalitis and encephalopathy of presumed infectious or autoimmune etiology. An altered mental state is a major criterion. Additional criteria (minor) to substantiate diagnosis include fever ≥38 °C (100.4 °F) within the 72 h before or after presentation; generalized or partial seizures not fully attributable to a pre-existing seizure disorder; new onset of focal neurological findings; cerebrospinal fluid (CSF) white blood count ≥5 mm^3^; and electroencephalographic abnormality that is consistent with encephalopathy and not caused by another factor or and not caused by another condition [30].

A clinical examination and a management plan for a child with encephalopathy should be developed concurrently. As soon as possible, a full history should be obtained. A thorough neurologic examination should be performed to localize brain damage and evaluate early prognostic indicators as well as to detect systemic symptoms such as rash, lymphadenopathy, and hepatosplenomegaly [31]. During a physical examination, clinical procedures such as mental status tests, memory tests, and coordination tests that record an altered mental state are commonly used to diagnose encephalopathy. Clinical test results are frequently used to diagnose or presumptively diagnose encephalopathy. When the altered mental state occurs associated with another primary disorder, such as chronic liver disease, kidney failure, anoxia, or a variety of other conditions, the diagnosis is typically made [29,32,33]. Glucose, ammonia, lactate, and ketone body levels in the blood as well as plasma acid–base status can all be used to help identify the subtype associated with genetic metabolic illnesses. The eventual diagnosis is based on certain laboratory findings at the start and/or during the static periods [2].

The cytokine storm subtype is distinguished by a significant increase in inflammatory tumor necrosis factor and interleukin concentrations in the serum and CSF [34]. Patients with disseminated intravascular coagulation and hemophagocytic syndrome have significant increases in ferritin, serum aminotransferase, pancreatic amylases, creatine kinase, creatinine, and uric acid nitrogen as well as ferritin, serum aminotransferase, pancreatic amylases, creatine kinase, creatinine, and uric acid nitrogen [33]. Clinical evidence, such as a biphasic pattern of seizure and varying degrees of altered states of consciousness as well as characteristic patterns of magnetic resonance imaging (MRI) and cerebral flow images using single-photon emission computed tomography, should be used to diagnose the excitotoxic crisis subtype [33].

EEG is a widely used technique for detecting and monitoring children with acute encephalopathy. Technological advancement has greatly simplified long-term bedside EEG monitoring. EEG has the advantage of being able to examine real-time brain function by recording electrical activity in the brain. Some children with acute encephalopathy are extremely ill and unstable in general. Even under these conditions, EEG monitoring is possible [2]. Several studies on long-term EEG monitoring among critically ill children with reduced consciousness, including those with acute encephalopathy, have recently been published. There have been numerous studies published on conventional EEG findings in children with acute encephalopathy. According to these results, EEG abnormalities are extremely common among children with acute encephalopathy. As a result, EEG is deemed to be useful in diagnosing acute encephalopathy. These EEG abnormalities include generalized/unilateral/focal slowness, low voltage, periodic lateralized epileptiform discharges, and paroxysmal discharges [35,36,37,38].

EEG has demonstrated its ability to detect nonconvulsive status epilepticus in AESD and FIRES/AERRPS (intermittent, latent seizures) [39,40]. EEG data may aid in differentiating AESD from long-term febrile seizures. Children with prolonged seizures and fever, reduced or absent spindles/fast waves as well as continuous or frequent slowing during sleep are diagnosed with AESD [41]. When combined with the clinical picture in patients with encephalopathy, EEG and brain imaging may improve diagnosis and have prognostic significance. The most common EEG finding in patients with encephalopathy is isolated persistent slowing of background activity. These patterns are linked to a variety of structural and non-structural pathologies.

The analysis of CSF is critical for determining the cause of encephalitis and distinguishing it from other types of encephalopathy. Lumbar puncture (LP) should be performed as soon as possible in suspected cases of encephalitis unless contraindicated. Clinical evaluation rather than cranial computerized tomography (CT) should be used to determine whether or not an LP is safe to perform [42,43]. Increased total protein and CSF/serum albumin quotient levels may be linked to severe edema [44]. Increased levels of cytokines and chemokines in CSF and serum may indicate an overly aggressive immune response [45]. CSF examination may reveal pleocytosis in some disorders [46], whereas pleocytosis may be uncommon in others [44].

Since 2000, imaging technology such as CT, MRI, SPECT, PET, and a variety of other neuroradiological tools have been used to treat heterogeneous acute encephalopathy syndrome. Acute encephalopathy was first defined using neuroradiographic images and clinical data derived from imaging, and it has since advanced significantly. It was possible to see fine cerebral edema images in acute encephalopathy. Figure 1, Figure 2, Figure 3 and Figure 4 illustrate imagining characteristics of MERS, ANE, AESD, and PRES, respectively. When acute encephalopathy is suspected, CT is usually the first test performed, because it is available in the majority of Japanese regional centers and has a quick imaging time. Acute encephalopathy is identified by cranial CT abnormalities [2,47], which include: (1) low-density zones spanning the entire brain or possibly the entire cerebral cortex, (2) no clear distinction between the cerebral cortex and the limbic system medulla, (3) both the surface of the cerebral subarachnoid space and the ventricles becoming narrower, (4) areas of low density: bilateral thalamus (ANE) and unilateral cerebral hemisphere (in some cases of AESD), (5) narrowing of the brain’s surrounding cisterns: swelling of the brainstem.

In some cases, a CT scan can be used to diagnose severe encephalopathy (for example, HUS encephalopathy), which has more edema in the brain than in mild cases [22]. MRI, in contrast, is a sensitive and non-radiological method for detecting encephalopathy, with diffusion-weighted imaging (DWI) being especially helpful in detecting early abnormalities. High-intensity lesions were either visible only on b = 3000 DWI for AESD, MERS, HSE, and unclassifiable encephalopathy or effectively identified on b = 3000 DWI than on b = 1000 DWI. Table 2 summarizes three infectious encephalopathy disorders for which neuroimaging is essential for diagnosis. The classifications of acute encephalopathy with febrile convulsive status epilepticus (AEFCSE), AIEF, and AESD are all part of a single spectrum and may refer to the same condition. On MRI diffusion-weighted images, a clinical form of AESD that specifically disrupts frontal lobe function in infants has been reported. As a result, the concept of AIEF is intended to be included in AESD: unlike AEFCSE, which has a biphasic course after a relatively short convulsive overlap, AESD has a biphasic course after a relatively short convulsive overlap, i.e., an initial febrile convulsive overlap followed days later by an afebrile partial convulsion with abnormal onset on MRI images. There is no English literature on AEFCSE. In Japan, there was controversy over whether AEFCSE, AIEF, and AESD were all the same disease; AIEF was a concept proposed based on cases of frontal lobe dominance with a course similar to AESD, whereas AEFCSE was a concept focused on the encephalopathy of the convulsive superimposed form of AESD. Since then, it has been determined that these three concepts are nearly identical to the AESD concept.

The biomarkers used to diagnose and assess the severity of acute encephalopathy differ depending on the type of encephalopathy (Table 3) [43,44,57,58].

Acute encephalopathy should be distinguished from other conditions that cause acute loss of consciousness during infectious diseases, such as intracranial infection (e.g., viral encephalitis and bacterial meningitis), autoimmune encephalitis, cerebrovascular diseases, traumatic, metabolic, and toxic disorders, and organ failure effects. The most recent Japanese guidelines listed several differential diagnoses of acute encephalopathy [2].

## 5. Management

The current national pediatric acute encephalopathy guideline [2] is based on expert consensus and case series and retrospective case-control studies for specific therapies such as corticosteroids [59], immunoglobulin [60], free-radical scavenger [61], osmotic agents [62], immunosuppressant [63], plasmapheresis, and therapeutic hypothermia [64]. Even though no drugs or therapeutic practices have been systematically demonstrated to lessen the sequelae of acute encephalopathy, the use of barbiturates and steroids has increased over time. This could be due to new research highlighting the importance of early aggressive therapy in the treatment of febrile status epilepticus [65]. Only surrogate markers such as fever and inflammatory changes in the CSF as well as neuroimaging are used to rule in or rule out infections in the early stages of infection. It is important to notice clinical clues from history and examination when narrowing down the etiology and deciding on an initial treatment approach. Proper head placement, suctioning of oropharyngeal secretions, and, if necessary, the use of oropharyngeal or nasopharyngeal airways should all be used to ensure airway patency in patients with diminished consciousness [42]. Children who exhibit signs of poor ventilation and oxygenation, such as irregular respiratory efforts, insufficient chest movements, poor air entry, central cyanosis, or peripheral oxygen saturation of 92% or less should be given a bag and mask first, which is followed by endotracheal intubation and mechanical ventilation. For emergency intubation, rapid sequence intubation is recommended to avoid aspiration and a rapid rise in ICP. Thiopental/midazolam, lidocaine, fentanyl, and a short-acting non-depolarizing neuromuscular blocking drug are among the induction agents (e.g., vecuronium, atracurium). Hypoglycemia and hyponatremia can accompany derangements in critically ill children, potentially exacerbating the underlying disease’s encephalopathy. When glucose levels of 60 mg/dL are treated promptly with 2 mL/kg of intravenous 25% dextrose, the neurologic symptoms are frequently reversed. Meanwhile, 5 mL/kg of 3% saline is required to raise sodium levels to acceptable levels in an asymptomatic child with a plasma sodium of 125 mEq/L [2].

Antimicrobials should be given to children with infectious diseases as soon as possible rather than waiting for laboratory confirmation.

Steroid pulse therapy, particularly cytokine storm therapy, is commonly used to treat virus-associated acute encephalopathy. The prognosis may be improved by beginning steroids within 24 h of the onset of ANE. It might be useful for treating encephalopathy caused by Escherichia coli O111, which produces Shiga toxin. However, in a recent study, steroid pulse treatment within 24 h did not improve the prognosis in children with suspected acute encephalopathy in the presence of AST. However, the authors noted that if treatment is started earlier, the neurological consequences of this illness could be avoided [66].

The cornerstone treatment for refractory status epilepticus is intravenous general anesthetics (such as midazolam, propofol, and barbiturates) [67]. General anesthetics, in contrast, can cause cardiovascular instability, respiratory suppression, infections, metabolic abnormalities, paralytic ileus, ischemic bowel, and thromboembolic events [68]. When general anesthetic therapy fails, several pharmacologic and nonpharmacologic approaches have been documented. Ketogenic diet can be considered for the treatment of acute encephalitis with refractory, repetitive partial seizures (FIRES/AERPPS). According to the findings of various case reports and retrospective studies, targeted temperature management can significantly improve the neurologic outcomes of acute encephalopathy [69,70,71,72,73]. Brain hypothermia therapy protocol has been established at hospitals for treating childhood status epilepticus and acute encephalopathy (Table 4) [73]. Hypothermia therapy may be useful in preventing the development of post-encephalopathic epilepsy (PEE) in the long-term consequences of AESD. Hypothermia therapy could help these patients enhance their quality of life by preventing the development of PEE [74]. Mild brain hypothermia therapy, followed by targeted temperature management, may be an effective way to improve neurological outcomes in children with HSES [75]. In contrast, there are also reports of inadequate benefits with brain hypothermia [76]. Classically, brain hypothermia was used to lower the body temperature from 32 to 34 °C. Although the effectiveness of this therapy has been suggested, the solution to side effects such as bradycardia, decreased blood pressure, and abnormal coagulation has been a problem. Therefore, in recent years, treatment for mild brain hypothermia that keeps the body temperature at 34.5 to 35.5 °C has been attracting attention as a new treatment strategy for pediatric acute encephalopathy. Table 5 summarizes the clinical manifestation, diagnosis, and treatment for major acute encephalopathy subtypes.

## 6. Acute Encephalopathy in the COVID-19 Era

Neurologic issues are common in COVID-19 patients who are hospitalized. Approximately 80% of hospitalized patients will experience neurologic symptoms at some point during their illness. According to recent studies, COVID-19-associated multisystem inflammatory syndrome in children (MIS-C) can cause cerebrovascular events as well as abnormal eye movements [90,91,92]. Neurological manifestations in patients with MIS-C were found in 27.1% in a recent study; 27% developed headaches, 17.1% developed meningism/meningitis, and 7.6% developed encephalopathy. Other uncommon neurological manifestations of MIS-C include anosmia, seizures, cerebellar ataxia, global proximal muscle weakness, and bulbar palsy. Neuroimaging revealed signal changes in the corpus callosum’s splenium in MIS-C patients with neurological features [93]. CSF pro-inflammatory chemokines and SARS-CoV-2 antibodies may serve as biomarkers of SARS-CoV-2 mediated NP-COVID-19 [94]. Electroencephalography revealed slow-wave patterns and nerve conduction studies, while electromyography revealed mild myopathic and neuropathic changes. Children with COVID-19 infection have been diagnosed with encephalitis, ANE, acute disseminated encephalomyelitis, cytotoxic lesion of the callosal splenium, posterior reversible encephalopathy syndrome, and other neurological illnesses [95]. Two methods have been postulated to explain how SARS-CoV-2 could cause neurological damage: a direct viral infection of the nervous system via ACE2 receptors and inflammatory harm caused by cytokine production [96]; in the latter instance, neurological symptoms could be a part of the overall picture. Acute encephalopathy research is still in its early stages. There have been case reports of anti-NMDA receptor encephalitis and ADEM in COVID-19 patients with severe psychotic symptoms [97,98,99]. Despite the absence of significant respiratory symptoms, neurological manifestations of COVID-19 infection are possible. Abnormal signals in the vast areas of the corpus callosum on MRI images have been reported in the case of a 5-year-old girl positive for COVID-19 [100]. Atypical manifestations in children, such as altered mental status and seizures as well as a hyperinflammatory shock with multiorgan dysfunction, should be noted by pediatricians. COVID-19 testing may be important in children with encephalopathy, as infected patients require extra measures to prevent further spread.

## 7. How to Evaluate Neurological Prognosis

In healthy developing infants, acute encephalopathy appears suddenly. Many cases necessitate rehabilitation training following a period of acute pediatric neurology or intensive care unit treatment. There have also been numerous cases of serious neurological sequelae. The question then becomes how long after the onset of the disease should a neurological prognosis be made and in what manner. According to recent reports, the timing of the assessment of neurological prognosis varies depending on the study design [66,74,79,101]. The Pediatric Cerebral Performance Category (PCPC) scale is the most commonly used scale for assessing disability in pediatric sequelae. The Pediatric Overall Performance Category (POPC) scale, the Wechsler Intelligence Test, and the Tanaka–Binay test are also commonly used. The PCPC scale is shown below (Table 6).

## 8. Conclusions

This review contains a plethora of current information on pediatric acute encephalopathies and sincerely thanks the efforts of the many outstanding researchers who have devoted their passion to the study and elucidation of pediatric acute encephalopathy.

## Figures and Tables

**Figure 1 jcm-11-05921-f001:**
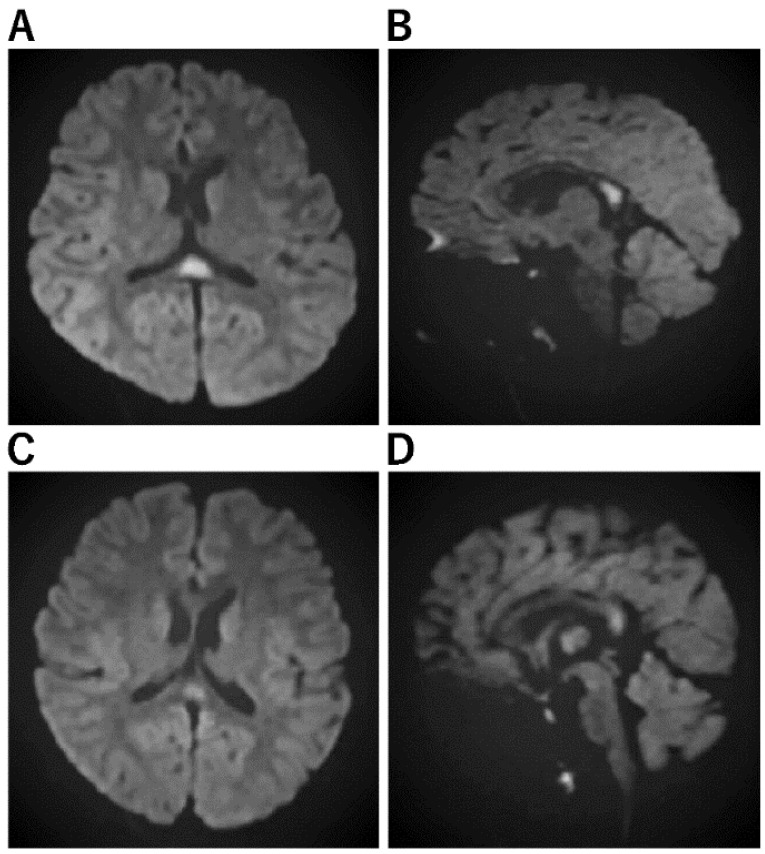
Imaging Characteristics of MERS. (**A**,**B**): A Horizontal/B sagittal section was performed on an 8-year-old boy who had an MRI on the third day of fever due to impaired consciousness and unable to recognize his own name. DWI showed an abnormal high signal in the cerebral corpus callosum: WBC 24800, CRP 7.84, Na 133, CL 95, ferritin 119.9, IL-6 171 EEG showed high amplitude slow waves in the occipital region. After 3 days of steroid pulse therapy, the fever resolved and consciousness improved. No sequelae. No causative organism or virus could be identified. (**C**,**D**): On the third day of vomiting and fever, he was hospitalized because he could no longer talk to his mother and could not look at her. He had diarrhea and was positive for rotavirus antigen in stool. In the bilateral frontal and occipital regions, EEG revealed persistent high amplitude slow waves. He was diagnosed with MERS on the fifth day after a diffusion-weighted MRI revealed an abnormally high signal in the corpus callosum. mPSL steroid pulse therapy was administered for three days, his level of consciousness improved, and both EEG and MRI were normalized.

**Figure 2 jcm-11-05921-f002:**
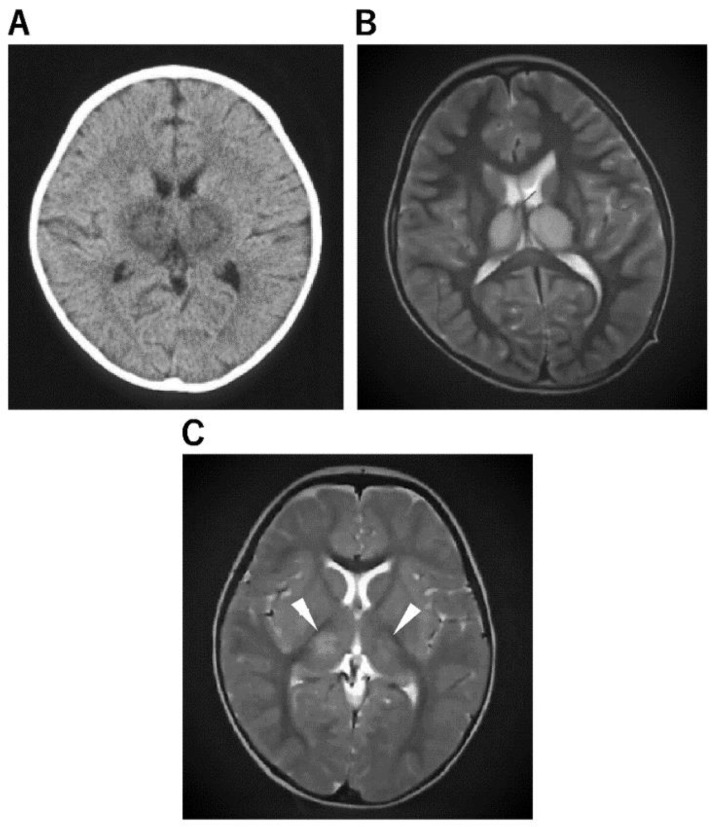
Imaging Characteristics of ANE. (**A**): An 11-month-old boy was admitted to the hospital after experiencing fever and vomiting. He was given a cold medicine prescription and sent home, but the next day, after a 3 min febrile convulsion, his loss of consciousness lasted 12 h, and a second 3 min convulsion was noted, so a CT was performed. He was diagnosed with ANE after a CT scan of the brain revealed abnormalities in the bilateral thalamus. (**B**): A 4-year-old boy visited the hospital with a high fever, vomiting, impaired consciousness, and convulsions. The rapid influenza A antigen test was positive, and MRI indicated abnormal signals in the bilateral thalamus not only on diffusion-weighted but also on T2 images, leading to the diagnosis of ANE. The patient was admitted to the intensive care unit immediately after being diagnosed with ANE. He was given cerebral sedation with high-dose barbital therapy and cerebral hypothermia at 34.5 °C for 48 h, which was followed by TTM as temperature control therapy, IVIG high-dose therapy, and mPSL steroid pulse therapy. Mitochondrial cocktail therapy was used in combination with 2 months after onset; the patient was able to walk, and 4 months later, his speech function had recovered to the same level as before the onset. (**C**): A 1-year and 2-month-old girl was admitted to the hospital with fever and partial seizures. After an MRI the next day, the T2-weighted image showed abnormal signals in the bilateral thalamus and diagnosed ANE. She was treated in the intensive care unit with 72 h 34.5 °C brain hypothermia, steroid pulse therapy, IVIG, and mitochondrial cocktail therapy. The patient was given cerebral sedation with high-dose barbital therapy and she was treated with dextromethorphan, which saved her life, but she was left with severe neurological sequelae.

**Figure 3 jcm-11-05921-f003:**
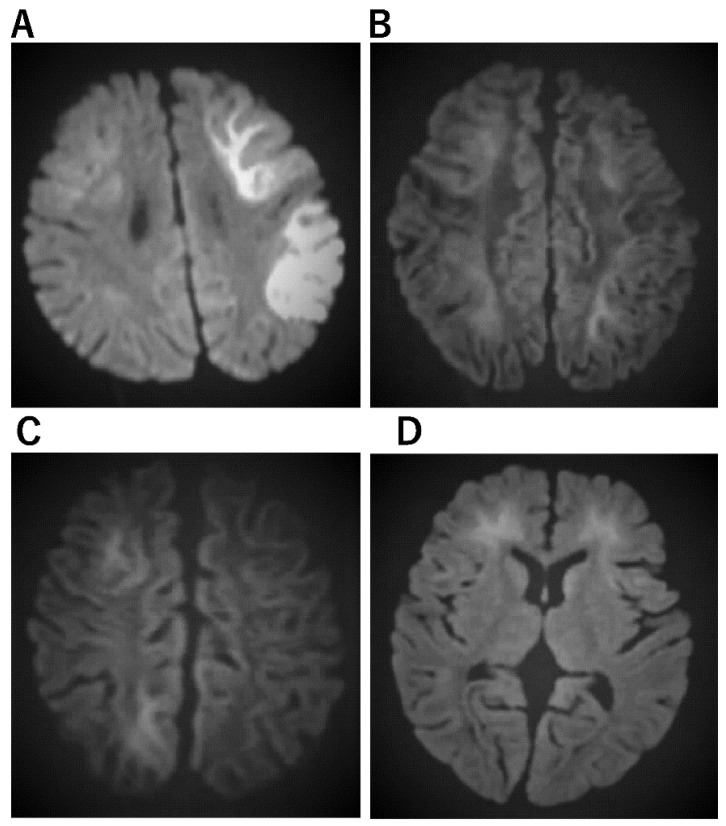
Imaging characteristics of AESD. (**A**) An 11-month-old boy who was admitted to the hospital with a high fever and 15 min seizure congestion of the right upper and lower extremities. The seizures stopped after the administration of midazolam. Thereafter, there was transient Todd’s palsy of the right upper and lower extremities. Brain MRI was normal. The fever resolved 3 days later and a rash appeared, which was clinically diagnosed as HHV-6 infection. The second diffusion-weighted brain MRI showed a bright tree appearance sign predominantly on the left side, diagnosing AESD. mPSL 30 mg/kg 3 days pulse therapy was administered. At the age of 6, he entered a regular elementary school, but his language skills were mildly poor. (**B**) A 3-year and 3-month-old girl. She has a 1 h febrile convulsion superimposed on fever. Midazolam brought the convulsions to a halt. The next day, she remained listless and was monitored with intravenous fluids; on the eighth day, she experienced a cluster of short convulsions in her limbs. Diffusion-weighted brain MRI revealed bilateral subcortical white matter predominance with bright tree appearance and an AESD diagnosis. Then, 48 h of mild cerebral hypothermia at 35.5 °C, steroid pulse therapy, and mitochondrial rescue therapy were performed. Six years after onset, she is living a normal fourth-grade elementary school life with no sequelae in terms of motor, language, or academic performance. (**C**) A 1-year and 7-month-old boy. After 4 days of febrile convulsive seizures, the fever subsided and a rash appeared; he was clinically diagnosed with HHV-6 infection. Multiple convulsive seizures lasting a few minutes were observed 5 days later. Slow waves were detected in the frontal and occipital regions of the EEG. Diffusion-weighted brain MRI showed an abnormally high signal in subcortical white matter and diagnosed AESD. mPSL pulse therapy and vitamin cocktail therapy were started. Body temperature was maintained at 35.5–36.0 TTM for 5 days The disease has been present for over two and a half years, and the child is now over 4 years old. There are no neurological sequelae and both language and motor functions are age-appropriate. (**D**) A 1-year-old boy with a fever of 39 °C and spontaneous convulsions that stopped spontaneously before reaching the hospital; 4 days later, he presents with two 3-min generalized convulsions and is rushed to the emergency room with no recovery of consciousness. He was admitted directly to the ICU, sedated with Rabonar, and given 48 h of mild cerebral hypothermia at 35 °C. Steroid pulse therapy was also administered. Thereafter, the temperature was kept at 36 °C, and the patient was transferred from the ICU to the general ward on the eighth day. On the same day, a brain MRI showed an abnormally high signal on diffusion-weighted images with bilateral frontal lobe predominance, and a diagnosis of AIEF-type AESD was made. Rehabilitation was continued until he was over 2 years old. After 1 year of onset, both his motor and language functions have recovered to the level of his age.

**Figure 4 jcm-11-05921-f004:**
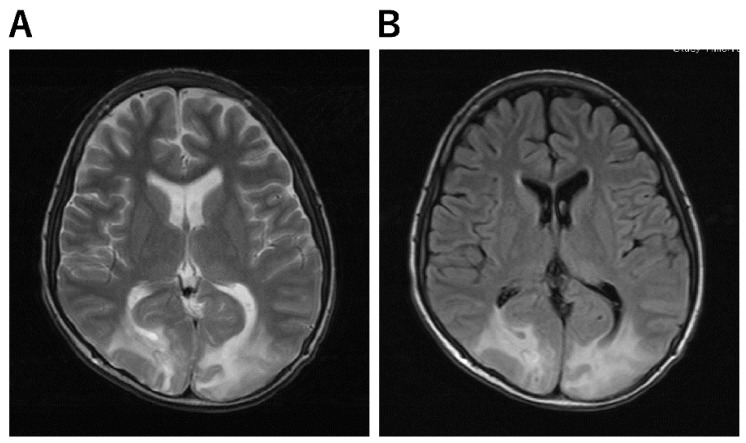
Imaging Characteristics of PRES. (**A**,**B**): A 13-year-old boy underwent skin graft surgery due to severe burns all over his body. He was on ventilatory management and sedatives for a long period of time postoperatively. As his generalized sepsis improved, his anesthetic was reduced and he was awakened; after 50 days, his consciousness improved completely; on day 51, he complained that “everything I see is white and I can’t see anything.” He then had a severe headache. His blood pressure was 150/89 mmHg and he had hypertension. Brain MRI scan showed an abnormal high signal in bilateral occipital areas on T2-weighted (**A**) and FLAIR (**B**) images, and he was diagnosed with PRES.

**Table 1 jcm-11-05921-t001:** Classification of Acute Encephalopathy [7,8,9,10,11,12,13,14,15,16,17,18,19,20,21].

Microbiological classification	• Influenza-associated encephalopathy • Human herpesvirus (HHV)-6/7 encephalopathy • Rotavirus encephalopathy • Respiratory syncytial virus encephalopathy • Herpes simplex virus encephalopathy • Varicella-zoster virus encephalopathy • Progressive multifocal leukoencephalopathies (PML) associated with the HIV virus such as subacute sclerosing panencephalitis (SSPE) caused by the measles virus and subacute encephalitis caused by the rubella virus. Bacterial infection-associated encephalopathy (Acute encephalopathy associated with hemolytic uremic syndrome (HUS) caused by *E. coli* O157:H7 and rotavirus infection and salmonella infection) [22] • Encephalopathy caused Bacillus cereulide-producing Bacillus cereus. • Mycoplasma infection-associated encephalopathy • Acute disseminated encephalomyelitis (ADEM) • Others
Metabolic errors	• Classic Reye syndrome • Encephalopathy secondary to inherited metabolic disorders (acute metabolic encephalopathy with carbamoyl phosphate synthetase 1 deficiency) [23]
Cytokine storm	• Encephalopathy with diffuse brain swelling Rey-like syndrome, sepsis-like encephalopathy) • Hemorrhagic shock and encephalopathy syndrome (HSES) • Acute necrotizing encephalopathy (ANE) • Non-herpetic limbic encephalitis (NHLE)
Excitotoxicity	• Acute encephalopathy with biphasic seizures and late reduced diffusion (AESD) • Acute infantile encephalopathy predominantly affects the frontal lobes (AIEF) • Hemiconvulsion–hemiplegiaepilepsy syndrome (HHE) • Anti-N-methyl-D-aspartate receptor encephalitis
Unknown or others	• Mild encephalitis/encephalopathy with a reversible splenial lesion (MERS) • Posterior reversible leukoencephalopathy syndrome (PRES or RPLS) [24] • Febrile infection-related epilepsy syndrome (FIRES) synonym: acute encephalitis with refractory, repetitive partial seizures (AERRPS) • Acute cerebellitis/cerebellopathy [25] • Epileptic encephalopathies with child onset • Acute encephalopathy with a background of genetic abnormalities in the early neonatal period (NEXMIF gene abnormality, Biallelic TBCD Mutations, mutations in ARX genes) [20,26] • Dravet syndrome • Acute encephalopathy associated with congenital adrenal hyperplasia (CAH) • Unclassified encephalopathy

**Table 2 jcm-11-05921-t002:** MRI/CT characteristics of acute encephalopathy.

Acute Encephalopathy Syndrome	Imaging Characteristics
Acute encephalopathy with biphasic seizures and late reduced diffusion (AESD) [29,48,49,50]	a. No abnormal lesion within 2 days. b. Subcortical white matter lesions between days 3 and 9, which are most obvious on DWI (bright tree appearance). The lesions are predominantly frontal or frontoparietal in a location with sparing of the peri-Rolandic region (central sparing). c. After 9 days, the bright tree appearance on DWI has disappeared, and T2WI or FLAIR imaging show high-intensity lesions with central sparing, and cerebral atrophy. d. MRS shows acute Glu elevation (days 1–4), changing to subacute Gln elevation (days 4–12). e. Cranial MRI in the late phase shows cerebral cortical lesions of reduced diffusion, indicating cellular edema of the subcortical white matter.
Acute infantile encephalopathy (AIEF) [51]	a. Postictal edematous changes in the white matter and cortex of both frontal lobes. b. Reduced perfusion in the frontal lobes occasionally persisted and was linked with increasing bilateral atrophic alterations and prolonged high signal intensity on T2-weighted images.
Acute encephalopathy with febrile convulsive status epilepticus (AEFCSE)	a. Bright tree appearance of subcortical white matter on MRI/DWI with central sulcus being spared. b. Residual atrophy in the frontal region, or frontal and parietal combined region on CT or MR.
Acute necrotizing encephalopathy (ANE) [52,53]	a. Concentric structure of the thalamocerebral lesions; diffuse cerebral edema and symmetric and multifocal lesions in the thalamus and other CNS regions, including the posterior limb of the internal capsule, posterior putamen, cerebral and cerebellar deep white matter, and upper brainstem tegmentum. b. The thalamic lesions often show hemorrhagic degeneration and cystic change after 3 days, showing a high signal on T1WI and a low signal on T2WI or T2 star-weighted imaging.
Clinically mild encephalitis/encephalopathy with a reversible splenial lesion (MERS) [54,55]	a. MRI-DWI shows abnormal signals in the vast portion of the corpus callosum. b. Abnormal intensities in the splenium on ADC-map, FLAIR, T1, and T2-weighted images.
Posterior reversible encephalopathy syndrome (PRES) [24,56]	a. Abnormal lesions in the occipital and parietal lobe areas. b. Extensive signal abnormality in the bilateral cerebellar hemispheres and within the thalami. c. T2-FLAIR images show signal abnormality within the midbrain, pons, and superior cerebellar peduncles. c. Fluid-sensitive MR sequences: parieto-occipital predominant white matter T2 hyperintensities

ADC: Apparent diffusion coefficient; CNS: Central nervous system; DWI: Diffusion-weighted imaging; FLAIR: Fluid attenuated inversion recovery; MRI: Magnetic resonance imaging; MRS: Magnetic resonance spectroscopy.

**Table 3 jcm-11-05921-t003:** Diagnostic markers for acute encephalopathy.

	Markers Indicating Poor Outcomes
Acute encephalopathy due to cytokine storm	Serum elevated aspartate aminotransferase Hyperglycemia Serum elevated TNF-α, cytochrome c Serum elevated IL-6, sTNFR1, IL- Serum elevated TIMP-1 Serum elevated soluble CD163 Serum elevated HMGB1 Hematuria or proteinuria
Acute encephalopathy due to excitotoxicity	Serum elevated MMP-9 Serum elevated MMP-9/TIMP-1 ratio CSF tau protein CSF S100B CSF VILIP-1 CSF IL-6 hsa-mir-34c hsa-mir-449b hsa-mir-449c serine/threonine kinase 39 gene

CSF: Cerebrospinal fluid; IL: Interleukins; HMGB1: High-mobility group box 1; MMP: Metalloproteinases; TIMP: Tissue inhibitors of MMP; TNF: Tumor necrosis factor.

**Table 4 jcm-11-05921-t004:** Proposed protocol for brain hypothermia therapy (Dokkyo Medical University Hospital ICU: April 2014) [73].

Anti-seizure medication treatment for status epilepticus (A) Midazolam (0.5 mg/kg) administered via the nasal cavity or cheek mucosa (B) Midazolam (0.15 mg/kg) intravenously (i.v.) administered (up to two doses possible) (C) Between ages 0 and 2 years: intravenous phenobarbital, between 15 and 20 mg/kg (10 min i.v.), ≥2 years: fosphenytoin 22.5 mg/kg (10 min i.v.) (D) Sodium thiopental between 3 and 5 mg/kg (slow i.v.)
Brain hypothermia therapy
This protocol applies to infants weighing ≥7.5 kg, and aged ≥6 months.
Introductory period 1. Status epilepticus/acute encephalopathy admission: ICU (request for admission), contact brainwave dept/radiology (brain and chest CT) 2. Check vital signs, establish a peripheral line 3. Establish central venous line: establish double/triple lumen catheter + arterial line 4. Fluid infusion between 80 and 100 mL/kg/day: under whole-body management, fluid control must not be reduced more than necessary to maintain blood pressure and cerebral circulation. Blood pressure is evaluated using an arterial pressure monitor. Maintenance fluids comprise an acetic acid preparation maintenance fluid and a lactic acid preparation. Vitamins are administered. When theophylline is administered, vitamin B6 is measured (light-shielding blood collection tube: administer vitamin B6 for theophylline-related seizures. Take care not to induce cardiac arrest by sudden administration of B6). 5. Management of blood count, electrolytes, blood sugars, albumin, and clotting value. Ferritin, IL-2R, β2MG, procalcitonin, immune globulin, etc. submitted. 6. Mannitol 3–5 mL/kg × 4–6 times/day (administered over 1 h). 7. Harvest spinal fluid (after the first administration of mannitol). General spinal fluid + various cytokines (IL-6, IL-1β, TNF-α), Tau protein, submitted. Freeze and store the remaining fluid at −80 °C. 8. If possible time-wise, implement MRI (DWI/ADC-map). 9. Intratracheal intubation (if difficult, use muscle relaxant or inhalation anesthetic) 10. Artificial ventilation: PCO_2_ at 35 to 40 mmHg (do not over-ventilate). PEEP kept slightly low considering brain hypertension. Raise head by 10°. If brain hypertension occurs: request placement of cerebral pressure monitor by a neurosurgeon. 11. Steroid pulse therapy: methylprednisolone 30 mg/kg for >2 h for 3 days, during which heparin or fragmin therapy is continued ≥APTT 1.5. 12. Administer famotidine 0.5 mg/kg twice, or omeprazole. 13. Brain hypothermia therapy: use a whole-body blanket-cooling method to induce target body temperature (direct intestine/bladder temperature of 34.0 to 35.0 °C) within 6 h of onset. If necessary, cool the head or wash the stomach with a normal saline solution while taking care not to cause electrolyte abnormalities, or use chilled fluid infusion. 14. Anti-seizure medication: sodium thiopental, 5–10 mg/kg/h (if this cannot be used, consider midazolam, 0.3 to 0.9 mg/kg/h). 15. Sedation depth should be confirmed by portable electroencephalograph/paperless electroencephalograph (Makin2) as reaching suppression burst within 6 h of beginning therapy. Cooling period 16. Target temperature to be maintained for 48 h (or a maximum of 72 h). Confirm the BIS monitor value at suppression burst (aim for 40 or below) and adjust the sodium thiopental dose administered based on the BIS value as appropriate. Cases achieving a positive sedation depth should have their sodium thiopental dose reduced prior to rewarming at a BIS value between 60 and 70 and at body temperature of 35.0 °C. Caution: If spikes remain with suppression bursts, consider complete suppression (pupils will constrict to mydriasis, and response to light is lost: BIS value of 20 or lower). 17. Use INVOSTM at an appropriate time to check oxygen saturation at the left and right front scalp to evaluate brain circulation. 18. Blood pressure maintenance: appropriate dose of dopamine hydrochloride (5 µg/kg/min = 0.3 mg/kg/h = 0.015 mL/kg/h), manage electrolyte abnormalities and blood glucose. Heart rate will fall to bradycardia with falling body temperature. 19. Administer antibacterial as appropriate. In applicable conditions, cerebroprotective edaravone, sivelestat Na as a neutrophil elastase inhibitor, and acyclovir. Rewarming period 20. Rewarming is implemented at a pace of 0.5 °C per 12 h. Care should be taken to avoid pneumonia in line with increased sputum secretions. Aim to remove the patient from artificial respiration on the fifth to seventh day. For cases in which laryngitis is likely, intravenous dexamethasone or epinephrine should be administered prior to removal of the tube. 21. For cases in which critical complications are envisaged, TRH therapy should be initiated at an early stage. 22. Including rehabilitation, aim to discharge the patient one month after onset. 23. Prior to discharge, evaluate brain waves, implement neuroradiological images/nuclear medicine tests and assess development. Where necessary, anti-seizure mediation should be periodically administered for preventative purposes.

**Table 5 jcm-11-05921-t005:** Acute Encephalopathy in Infancy and Childhood.

Subtype Description	Incidence in Japan	Clinical Manifestations	Diagnosis	Treatment
ADEM [2,77]				
An immune-mediated inflammatory demyelinating condition that predominately affects the white matter of the brain and spinal cord.	0.40 per 100,000	Polyfocal neurologic deficits and is typically self-limiting.	Based on clinical features and findings on neuroimaging and laboratory investigations. ADEM lacks a specific identified biological marker rendering a reliable laboratory diagnosis, long-term follow-up is important as there are instances where an illness initially diagnosed as ADEM is ultimately replaced with a diagnosis of MS	Combination of intravenous corticosteroids and IVIG, (2) cyclosporin, (3) cyclophosphamide, or (4) plasma exchange/plasmapheresis
AESD [78,79]				
Biphasic seizure and altered consciousness during the acute phase, followed by restricted diffusion in bilateral cerebral parenchyma on MRI during the subacute stage	Incidence is higher in Asian countries, especially Japan, and the genetic background may be a possible higher incidence	A prolonged febrile seizure is the first symptom. Brief seizures may be present in mild cases. Involuntary movements may act prognostic factor AESD prediction score by Tada et al. [80] Level of consciousness 12–24 h after seizure GCS 14–13 (JCS 1–3) and GCS 12–9 (JCS 10–30) were scored 2, GCS 8–3 (JCS 100–300) scored 3, age below 1.5 years scored 1, duration of ES above 40 min scored 1, mechanical intubation scored 1, AST on admission above 40 mEq/l scored 1, blood glucose on admission above 200 mg/dl scored 1, and Cr on admission above 0.35 scored 1 AESD prediction score by Yokochi et al. [81] pH < 7.014 1 point ALT ≥ 28 2 points Glu ≥ 228 2 points Time to awakening ≥ 11 h 2 points Cre ≥ 0.3 1 point Ammonia ≥ 125 2 points 4 points or more: High-risk group for AESD for both scores [77,78]	High signal intensity on DWI for the anterior area Severe acidosis is the most common lab finding. EEG may usefully differentiate AESD from PFS	Cyclosporine, methylprednisolone pulse therapy, intravenous immunoglobulin, and other therapies that suppress inflammatory cytokines. TTM may be considered but need to differentiate AESD from FSE at the early acute stage. TTM during the prolonged convulsive phase, prior to the diagnosis of AESD, could prevent the patient from developing a second phase of convulsions and thus prevent the patient from developing AESD.
AEFCSE				
Develops with prolonged febrile convulsion, followed by mild unconsciousness, then subsequently provoking a cluster of convulsions (late seizures) with a comatose state.	Incidence is higher in Asian countries, especially Japan	Pyrexia followed by partial seizures and subsequently hemiconvulsions. Transient neurological symptoms and Intellectual disability, attention deficit	EEG findings showed slow waves predominantly on the right hemisphere and a high signal in the subcortical white matter as with the cerebral lobar distribution pattern. Acute infantile encephalopathy predominantly affects the frontal lobes (AIEF). Thermolabile genotype of CPT II variations consisting of three single nucleotide polymorphisms in exons 4 (1055T > G/F352C and 1102G > A/V368I) and 5 (1939A > G/M647V)	TTM
ANE [82,83,84]				
Multiple bilateral brain lesions, primarily involving the thalami, but also involving the putamina, internal and external capsules, cerebellar white matter, and the brainstem tegmentum.	Many cases have been reported in Asia as well as in a number of Western countries.	Dramatic neurological deficits/symptoms. Neurological deficits may be preceded by a viral prodrome.	Associated thalamic, putamina, cerebral, cerebellar, and brainstem abnormalities are hypodense on CT. Bilateral symmetrical thalamic involvement. Abnormal signals on MRI are hypointense on T1 and hyperintense on T2. Restricted diffusion of the involved regions. ANE of childhood can be distinguished from ADEM clinically by the onset of encephalitic features shortly after the prodromal illness, whereas in ADEM, they may take 1 to 2 weeks to develop.	Immunomodulatory therapy such as corticosteroids or intravenous immunoglobulin is often used. TTM has also been used. TTM is critical to the outcome of children with ANE, especially if started within 12 h of onset.
MERS [85,86,87]				
An infection-associated encephalitis/encephalopathy syndrome that is predominately caused by a virus or a variety of pathological conditions, including MIS-C. There have been scattered reports of MERS associated with AFBN and Kawasaki disease. AFBN + MERS, a urinary tract infection in children, is particularly noteworthy		Fever, headache, neck rigidity, and Kerning sign (+)	serum VCA IgG (+), EBNA-1 IgG (−), EBV IgM (−), and inflammation in the analysis of CSF Cranial MRI+C showed that the blood vessels on the surface of the brain were increasing and thickening, and diffuse slow waves were detected on the EEG The high-signal intensity in the splenium of the corpus callosum on T2W Splenial hyperintensity as a “boomerang sign” on DWI and reduced diffusion on ADC	Supplementation, steroids and IVIG, acyclovir, and prescribed oral sodium, but some cases improve with the natural course of the disease.
PRES [56]				
A clinical–radiological syndrome characterized by a headache, seizures, altered mental status, visual loss and white matter vasogenic edema affecting the posterior occipital and parietal lobes of the brain predominantly.		Visual disturbance can vary from blurred vision and homonymous hemianopsia to cortical blindness. Altered consciousness may vary from mild confusion or agitation to coma.	The bilateral occipital, parietal, frontal cortex, and subcortical white matter T2/fluid-attenuated inversion recovery hyperintensities	Intravenous fluids, antibiotics, antiepileptics
FIRES [88,89]				
A subtype of NORSE that requires a prior febrile infection between 2 weeks and 24 h before the onset of refractory status epilepticus with or without fever at the onset of status epilepticus.	Very rare, affects approximately 1 in a million children	Focal seizures with impaired awareness and bilateral tonic–clonic seizures. Seizures progress to continuous or nearly continuous seizures.	EEG Leptomeningeal enhancement, bilateral claustrum hyperintensity or progressive mesial temporal lobe atrophy on MRI.	Benzodiazepines, Barbiturates, Ketamine, Lidocaine, Magnesium, Ketogenic diet, Cannabidiol

ADEM: Acute disseminated encephalomyelitis; ANE: Acute necrotizing encephalopathy; AEFCSE: Acute encephalopathy with febrile convulsive status epilepticus; AESD: Acute encephalopathy with biphasic seizures and late reduced diffusion; EEG: Electroencephalogram; CSF: Cerebrospinal fluid; DWI: Diffusion-weighted imaging; FIRES: Febrile infection-related epilepsy syndrome; FLAIR: Fluid attenuated inversion recovery; MERS: Mild encephalitis/encephalopathy with a reversible splenial lesion; MRI: Magnetic resonance imaging; PRES: Posterior reversible leukoencephalopathy syndrome; TTM: Targeted temperature management.

**Table 6 jcm-11-05921-t006:** PCPC Scale [102].

Score	Category	Description
1	Normal	At an age-appropriate level; school-age children attend regular school
2	Mild disability	Conscious, alert, able to interact at age-appropriate level; regular school, but grades perhaps not age-appropriate, possibility of mild neurologic deficit
3	Moderate disability	Conscious, age-appropriate independent activities of daily life; special education classroom and/or learning deficit present
4	Severe disability	Conscious, dependent on others for daily support because of impaired brain function
5	Coma or vegetative state	Any degree of coma, unaware, even if awake in appearance, without interaction with the environment; no evidence of cortex function; possibility for some reflexive response, spontaneous eye-opening, sleep–wake cycles
6	Brain death/death	Brain death, death

PCPC: Pediatric Cerebral Performance Category.

## Data Availability

Not applicable.

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
