# Peer review of "A Comprehensive Review of Pediatric Acute Encephalopathy"

_jcm, 2022, doi:10.3390/jcm11195921_

Round 1

Reviewer 1 Report

An extensive review on  somewhat confusing and overlapping entities. The authors have described in extensive texts with somewhat overlap with the accompanying tables. Would request them to reduce the descriptive texts wherever the same thing is described in the tables to avoid repeatations and monotony.

Some English language corrections are required. For example, in the Abstract ' Acute Encephalitis in children is the result of our predecessors zealous pursuit...'. Needs to be corrected.

In Management it's written ' Within 24 h,  steroid pulse treatment did not improve the prognosis with suspected acute encephalopathy and increased AST. Even if treatment is started earlier, the neurological consequences maybe avoided.' This line needs rectification.  It maybe possible to reduce the text in Management section, as detailed descriptions are provided in TABLE 5 and 6. These are the minor changes I had alluded to earlier.

Author Response

Response: Thank you for your review and we appreciate your remarks and suggestions. We have now corrected the English language throughout the document and also fixed statements for better clarity. We have now reduced the text in the management section as per your suggestion.

Reviewer 2 Report

This review describes the current state-of-art of  pediatric acute encephalitis/encephalopathy, with particular regards to diagnosis and management.

Evident efforts have been made to cover, in a comprehensive way, this argument considering the extensive literature in this field. The paper is globally interesting and well written and could be useful to give to readers a clear and concise description of the state of art of this topic.

I think that a methods section, in which authors have to provide on which Online database the Literature review was conducted and define the years range of search and inclusion/exclusion criteria, is needed. In addition, also a flow diagram would be helpful.
Define a discussion section after the introduction describing the results of the literature search and then divide it in sub sections.

Please check carefully the paper since there are some minor mistakes (e.g. PERS instead of PRES).

Author Response

Response: We welcome the comments and suggestions you have provided. We have now included the details of the literature search in the revised manuscript. The original search was limited by language, population age, and year of publication. From the original hits, we included the relevant articles based on manual screening.  Since there are very few steps in this approach, we have not provided the algorithm. 

We have now corrected the abbreviation for PERS throughout the manuscript.

Reviewer 3 Report

Thank you for the opportunity of reviewing this manuscript. The author aims to provide a comprehensive review of paediatric encephalitis/encephalopathy. Unfortunately, I think there are several major issues with the manuscript.

Firstly, I thought that the flow of text was extremely poor throughout, and I had difficulty following the concepts presented. For example, in the final paragraph of section 1, the author discusses the differential diagnosis of encephalopathy, however then goes on to discuss meningitis in detail, including testing for pathogens via latex kits and PCRs. I don't think this is particularly relevant. There are many similar jarring transitions throughout the rest of the text.

Secondly, I don't think that the differences between encephalopathy and encephalitis have been appropriately highlighted/discussed. The title of the manuscript indicates a review of both conditions, however the majority of the text discusses encephalopathy. Concerningly, there are some instances where encephalitis and encephalopathy are interchanged. For example, Table 2 supposedly provides the diagnostic criteria for encephalopathy. In reality, the criteria provided are for encephalitis. It is crucial that the author clearly delineates the scope of the article.

Some sentences appear to be duplicated. For example, in Section 1, Paragraph 1, the phrase 'subsyndromal delirium, delirium, or coma' appears twice. In Section 5, Paragraph 1, the sentence 'only surrogate markers such as fever and inflammatory changes in the CSF as well as neuroimaging, are used to rule in or out infections in the early stages of infection' appears twice in a row.

There appears to be quite a lot of superfluous/misplaced text. For example, Section 3 is on clinical presentation, but half of the first paragraph describes pathophysiology. Section 5 discusses management, but Paragraph 4 starts of with discussing ketogenic diets and cannabinoid treatment for epilepsy.

In summary, I think there are some major issues with the manuscript, including the ill-defined scope, interchanging of two very different conditions (encephalopathy vs encephalitis), and poor readability.

Author Response

Thank you for the opportunity of reviewing this manuscript. The author aims to provide a comprehensive review of paediatric encephalitis/encephalopathy. Unfortunately, I think there are several major issues with the manuscript.

Response: We appreciate you reading our manuscript and making insightful recommendations.

Firstly, I thought that the flow of text was extremely poor throughout, and I had difficulty following the concepts presented. For example, in the final paragraph of section 1, the author discusses the differential diagnosis of encephalopathy, however then goes on to discuss meningitis in detail, including testing for pathogens via latex kits and PCRs. I don't think this is particularly relevant. There are many similar jarring transitions throughout the rest of the text.

Response: Thank you for your comments. We have now revised the text for better flow throughout the manuscript and revised statements for better clarity.

Secondly, I don't think that the differences between encephalopathy and encephalitis have been appropriately highlighted/discussed. The title of the manuscript indicates a review of both conditions, however the majority of the text discusses encephalopathy. Concerningly, there are some instances where encephalitis and encephalopathy are interchanged. For example, Table 2 supposedly provides the diagnostic criteria for encephalopathy. In reality, the criteria provided are for encephalitis. It is crucial that the author clearly delineates the scope of the article.

Response: Thank you for your comments. We have now removed the text related to encephalitis in the revised manuscript. Regarding Table2, because of the significant clinical overlap between encephalitis (infectious and noninfectious) and encephalopathy of presumed infectious etiology, the case definition is formulated to capture both syndromes by the International Encephalitis Consortium.

We have now revised the table heading to “The International Encephalitis Consortium recommends diagnosis of encephalitis and encephalopathy of presumed infectious or autoimmune etiology acute encephalopathy be based on the following criteria:”

Some sentences appear to be duplicated. For example, in Section 1, Paragraph 1, the phrase 'subsyndromal delirium, delirium, or coma' appears twice. In Section 5, Paragraph 1, the sentence 'only surrogate markers such as fever and inflammatory changes in the CSF as well as neuroimaging, are used to rule in or out infections in the early stages of infection' appears twice in a row.

Response: Thank you for your comments. We have now removed, and revised statements for better clarity.

There appears to be quite a lot of superfluous/misplaced text. For example, Section 3 is on clinical presentation, but half of the first paragraph describes pathophysiology. Section 5 discusses management, but Paragraph 4 starts of with discussing ketogenic diets and cannabinoid treatment for epilepsy.

In summary, I think there are some major issues with the manuscript, including the ill-defined scope, interchanging of two very different conditions (encephalopathy vs encephalitis), and poor readability.

Response: Thank you for your comments. We have now revised the text for better flow throughout the manuscript and removed, and revised statements for better clarity.

Round 2

Reviewer 3 Report

Thanks for submitting your manuscript with corrections. I think it is more readable now and many of my concerns have been addressed! There are some minor typos throughout the text which can be fixed in final editing.